Effects of video game immersion and task interference on cognitive performance: a study on immediate and delayed recall and recognition accuracy

http://orcid.org/0000-0002-9837-526X Mancone Stefania
http://orcid.org/0009-0002-6273-6163 Tosti Beatrice
http://orcid.org/0000-0003-2064-3372 Corrado Stefano
http://orcid.org/0000-0002-5470-3233 Diotaiuti Pierluigi p.diotaiuti@unicas.it
Department of Human Sciences, Society and Health, University of Cassino and Southern Lazio , Cassino, Lazio , Italy
Epifania Ottavia M.
Electronic publication date: 2024 Oct 9
Publication date: 2024
Volume: 12
Electronic Location ID: e18195
Received 2024 May 24; Accepted 2024 Sep 9
Copyright: © 2024 Mancone et al.
Copyright year: 2024
Copyright holder: Mancone et al.
License: This is an open access article distributed under the terms of the Creative Commons Attribution License, which permits unrestricted use, distribution, reproduction and adaptation in any medium and for any purpose provided that it is properly attributed. For attribution, the original author(s), title, publication source (PeerJ) and either DOI or URL of the article must be cited.
License URL: https://creativecommons.org/licenses/by/4.0/

Keywords: Multitasking, Cognitive interference, Task switching, Memory recall, Digital game immersion, Cognitive load, Error analysis, Memory recognition, Attentional resources, Human-computer interaction

Funding: The authors received no funding for this work.

==============================
This study investigates the cognitive impacts of video game immersion and task interference on immediate and delayed recall as well as recognition tasks. We enrolled 160 subjects aged 18 to 29, who were regular players of “shoot-em-up” video games for at least 3 years. Participants were assigned to one of three experimental groups or a control group. The experimental conditions varied in the timing and type of tasks: the first group performed a video game session between recall tasks, the second group multitasked with video games and recall tasks simultaneously, and the third group engaged in task switching from video games to recall tasks. Using the Rey Auditory Verbal Learning Test, we measured the effects of these conditions on cognitive performance, focusing on error types and recall accuracy. Results indicated that multitasking and task switching significantly affected the subjects’ performance, with notable decrements in recall and recognition accuracy in conditions of high task interference. The study highlights the cognitive costs associated with multitasking in immersive digital games and provides insights into how task similarity and interference might increase error rates and affect memory performance.

Introduction

The human capacity for multitasking has long been a focus of psychological research, with studies often revealing a trade-off between task performance and the number of tasks being managed simultaneously (Kahneman, 1973; Pashler, 2000). This trade-off is particularly evident in situations where tasks compete for similar cognitive resources, leading to what is commonly referred to as “cognitive interference” (Allport, Antonis & Reynolds, 1972). While the ability to multitask is frequently touted in everyday life, empirical evidence suggests that performance suffers when individuals engage in multiple tasks, especially those that are similar in nature (Brooks, 1968; Kahneman & Treisman, 1984).

The advent of digital technologies has heightened the prevalence and complexity of multitasking environments, with video games representing a particularly immersive form of digital multitasking. Video games, particularly action genres like first-person shooters (FPS), demand continuous attention, rapid response, and complex multitasking skills, making them an ideal medium to study the effects of immersion and task interference on cognitive functions (Green & Bavelier, 2006). The increasing complexity of modern life demands a deeper understanding of how simultaneous activities, particularly immersive digital tasks, impact cognitive tasks such as recall and recognition.

Recent literature suggests that the immersive nature of video gaming can significantly modulate cognitive performance, particularly when gaming is coupled with other cognitive tasks (Strobach, Frensch & Schubert, 2012; Boot et al., 2008). Literature has delved into the impact of video gaming on cognitive performance, particularly when coupled with other cognitive tasks. While some studies have suggested that video game training may not significantly enhance domain-general cognitive abilities (Sala, Tatlidil & Gobet, 2018), others have highlighted the potential for action video games to enhance attentional control (Bavelier & Green, 2019). The application of video games for cognitive and emotional training in the adult population has been systematically reviewed, with findings indicating that puzzle games may not necessarily enhance cognitive skills compared to other video game genres (Pallavicini, Ferrari & Mantovani, 2018). Furthermore, age-related differences in video gaming have been suggested to potentially influence cognitive function enhancement in younger adults more than in older adults (Choi et al., 2020).

Literature has explored the cognitive abilities displayed by action video gamers and non-gamers, with evidence suggesting that video game playing may improve selective attention and speed, while also potentially causing an indirect decline in these abilities by promoting a sedentary lifestyle (Kowal et al., 2018; Özçetin et al., 2019). Also the immersive nature of video gaming has been emphasized, with studies indicating that video gaming offers an immersive experience distinct from other forms of media engagement (Pietersen et al., 2019). The potential for video gaming to affect cognitive functions and their neural bases in adolescents and young adults has been investigated, with results suggesting that video gaming may enhance brain areas such as the hippocampus (Alho, Moisala & Salmela-Aro, 2022).

The impact of video game-based interventions on cognitive function in older adults has been evaluated, with findings suggesting that video gaming has the potential to improve cognitive flexibility and task switching paradigms (Tartar & Hewlings, 2019). The cognitive abilities of action video game and role-playing video game players have been compared, with evidence indicating that action video game play has been positively associated with tasks requiring cognitive flexibility, such as multitasking and working memory (Bavelier, Bediou & Green, 2018). The potential for video games to modulate cognitive factors related to immersion has been highlighted, potentially leading to excessive practice and impulsive behavior (Rémond & Romo, 2018). The integration of immersive virtual reality (VR) methods in neuropsychology has recently garnered attention for its potential to closely mimic real-world scenarios, thereby enhancing the ecological validity of cognitive assessments. VR offers unique opportunities to create highly controlled, immersive environments for cognitive assessments, providing a level of ecological validity that traditional methods often lack (Kourtesis & MacPherson, 2021). However, there remains a gap in understanding the specific nature of these effects—especially the types of errors induced by such multitasking scenarios and how task similarity exacerbates interference.

Recent studies have delved into the detrimental effects of multitasking on cognitive performance and the neural adaptations resulting from extensive multitasking training (Wiradhany & Koerts, 2019). These adaptations may modulate the extent of cognitive interference experienced by individuals, suggesting a potential area for further investigation into how habitual exposure to multitasking environments could impact cognitive resilience or vulnerability to interference (Beuckels et al., 2019). The correlates of habitual media multitasking behavior have been a subject of increased study, with recent research distinguishing heavy from light media multitaskers (Parry, Roux & Bantjes, 2020). Media multitasking has been associated with adverse cognitive, psychosocial, and functional outcomes, emphasizing the need for further exploration into its effects (Kokoç, 2021). Attention control has been identified as playing a critical role in mitigating the negative effects of multitasking with social media on academic performance among adolescents (Zaremoodi & Haffari, 2019).

This study is framed within the dual-process theory, which distinguishes between automatic and controlled cognitive processes (Shiffrin & Schneider, 1977). FPS games were selected for this study due to their well-documented impact on visual attention, spatial reasoning, and executive control, which are cognitive domains closely related to the tasks being assessed. By focusing on these genres, the study aims to explore how the high cognitive load imposed by these games interacts with memory and recognition tasks, offering insights into the broader implications of digital multitasking. We hypothesize that immersion in video games prior to or during other cognitive tasks will negatively affect performance, increase error rates, and impact task switching efficiency. This research not only contributes to the theoretical framework of cognitive multitasking but also has practical implications for educational strategies and our understanding of cognitive load management in digital environments.

Materials and Methods

Participants

We recruited 160 participants aged between 18 and 29 years, all of whom had at least 3 years of regular experience playing “shoot-em-up” video games. Participants were recruited from a university setting (67% male, 33% female) across various academic majors, resulting in a diverse sample with different cognitive and gaming backgrounds. Participants were randomly assigned to one of three experimental groups or a control group using a computer-generated randomization algorithm. This process was designed to ensure that each group was balanced in terms of age, educational background, and gaming experience. The randomization was stratified to maintain an even distribution of gender across all groups. This approach ensured that any differences observed in cognitive performance could be attributed to the experimental conditions rather than demographic imbalances. Recruitment was conducted on a voluntary basis, with participants fully informed and providing consent prior to their involvement in the study. Participants were primarily recruited through targeted announcements in psychology courses and by word of mouth among students attending various courses (Economics, Law, Educational Sciences, Engineering, Classical Studies, Sports Science, Social Work). We opted for presentations at the beginning or end of lectures to explain the study’s purpose, its relevance to understanding cognitive processes in gaming, and how students could volunteer. Additionally, students already enrolled were encouraged to inform their peers attending other majors about the study, leveraging peer networks to increase participant numbers. No incentives were provided for participation.

Design and procedure

The study employed a between-subjects design to examine the effects of video game immersion and task interference on cognitive performance. Each experimental condition was designed to vary the type of cognitive interference experienced by participants while engaging in a secondary task of mnestic recall and linguistic recognition.

Criteria for game selection

The game genre was restricted to first-person shooters (FPS), a genre well-known for its demand on attention, speed, and multitasking abilities. These games are highly popular among the age group of our participants (18–29 years), which ensures that the majority of participants are already familiar with the game mechanics, reducing the learning curve and focusing on the cognitive load aspect. The study utilized “Call of Duty: Modern Warfare”, a first-person shooter (FPS) game, chosen for its high cognitive demands, popularity among the target demographic, and ability to impose a significant cognitive load. This selection was based on pre-testing and literature review, ensuring the game’s suitability for examining the effects of task interference on cognitive performance. The game’s difficulty level was configured to be medium-high to ensure it was sufficiently challenging to affect cognitive performance without being overly frustrating for the participants.

Implementation

During the study, the game sessions were standardized across all participants to control for any variability in game difficulty and session duration. Participants played the game under monitored conditions to ensure consistent exposure and to accurately measure the game’s impact on subsequent cognitive task performance.

The Rey Auditory Verbal Learning Test (RAVLT) (Rey, 1958; Carlesimo et al., 1996) was chosen as the primary tool for assessing immediate and delayed recall as well as recognition memory. The test is made up of seven tasks. In the first task the examiner reads the words and asks the subject to repeat them. After the first task, the words are read another four times and the subject is asked to repeat them, including the previously recited words. After these five tasks, the subject under examination is “distracted” with visual-spatial tasks for about 15 min, and then he is asked to repeat the same words one more time. The subject under examination then undergoes another recognition task whereby he is presented verbally with 46 words, that is to say, the 15 words presented to him in the first task together with 31 distractors. They are asked to identify the words they remember from the original list. The test serves as a classic quantitative evaluation for immediate and deferred recall, but also for an interesting qualitative evaluation on how the subject has coped with the memorization test.

The subjects were randomly assigned to one of three experimental groups. The first group completed a 15-min gaming session in a familiar environment with a medium-high level of technical difficulty during the interval in the administration of the Rey test, i.e., between the first and the second session. The subjects were asked to focus on the video game task and simultaneously follow the verbal instructions of the experimenter before and after the gaming session. The second group was invited to play a video game while simultaneously responding to the verbal prompts of the experimenter who administered the first session of the Rey test (immediate recall of a list of fifteen words in five repeated phases). At the end of the fifth repetition, there was a 15-min interval, as prescribed by the test, and then we proceeded to complete the second part of the test that measured the deferred recognition of the preceding list from a new list of 46 words. The experimenters recorded the number of correct recalls and also noted the number and type of errors made during the recall and recognition sessions. The third group was first invited to play for 15 min and then to undertake the recall and recognition test sessions. Overall, an experimental structure was established in which the first group experienced an automated immersion in the video game activity between one test and the other; the second group operated under conditions of maximum interference and simultaneity between the automated video game activity and the mnestic recall; the third group performed the mnestic task immediately after the automated immersion. In addition to the experimental groups, we used another control group consisting of 40 subjects from the same age range (18–29) to whom we administered the Rey test without any experimental game conditions. All experimental sessions were conducted in soundproof rooms with standardized lighting conditions to minimize external distractions and ensure consistency across sessions. Experimenters were blinded to the group assignments to prevent bias in data collection and analysis.

Summary of experimental conditions

Group 1) Participants performed the RAVLT, followed by a 15-min video game session, and then completed the recall and recognition tasks.

Group 2) Participants engaged in video gaming simultaneously while trying to recall the words from the List, immediately followed by the recognition task.

Group 3) Participants played video games for 15 min before starting the RAVLT.

Control Group: Participants only completed the RAVLT without any interference from video gaming.

Confidentiality regarding the data collected was strictly maintained, with measures іn place tо protect the privacy and anonymity оf all participants. The study was conducted in accordance with the requirements of the Helsinki convention and approved by the local Institutional Review Board of the University of Cassino and Southern Lazio (IRB_DIPSUS 07:22/04/23). Informed written consent was obtained from all participants.

Data analysis

The performance on the RAVLT was quantified by the number of words correctly recalled during the immediate and delayed recall tasks, as well as the number of correctly recognized words in the recognition task. Differences between groups were analyzed using ANOVA to assess the impact of task interference on recall and recognition. Error types (intrusions and repetitions) were also recorded and analyzed to understand the patterns of errors under different multitasking conditions. In addition to standard ANOVA, interaction effects between task types and group assignments were analyzed to discern more nuanced impacts of task sequencing and multitasking on cognitive outcomes. These analyses help in understanding whether the sequence in which tasks are performed (i.e., before or during video game play) exacerbates or mitigates the cognitive load, potentially influencing the recall and recognition performance differently.

The level of statistical significance was set at p < 0.05. In order to ensure the robustness of our findings, a power analysis with G*Power (version 3.1.9.7) was conducted for the primary statistical tests (ANOVA) used in this study. Given the sample size of 160 participants distributed across four groups, the power analysis indicated that the study had a power of 0.95 or higher to detect medium to large effect sizes (Cohen’s f > 0.25) at an alpha level of 0.05. This high level of statistical power suggests that the study is well-equipped to identify significant differences between groups, minimizing the risk of Type II errors. To address potential overfitting, we employed k-fold cross-validation techniques, where the dataset was divided into k subsets, and the model was trained and validated k times, each time using a different subset as the validation set and the remaining subsets as the training set. This approach helps ensure that the model’s performance is more robust and generalizable, reducing the likelihood that the model is simply fitting noise in the training data. Missing data were handled using multiple imputation for cases where less than 5% of data was missing. For cases with more extensive missing data, those records were excluded from the analysis to prevent bias. Outliers were identified using z-scores and were handled by winsorizing the data at the 5th and 95th percentiles. This approach allowed us to reduce the influence of extreme values while retaining all data points for analysis. The impact of outliers on the overall results was assessed, and sensitivity analyses were conducted to ensure the robustness of the findings. Data analyses were performed using the IBM SPSS Statistics 26.

Results

The study investigated the effects of video game immersion and task interference on cognitive functions, specifically focusing on immediate and delayed recall and recognition tasks. Statistical analyses revealed significant differences in cognitive performance across different conditions of task interference. Table 1 reported below includes mean values, standard deviations, and sample sizes for each group, providing a quick visual comparison across the different experimental conditions.

Table 1 Comparative analysis of cognitive performance across experimental groups.

Group	Immediate recall mean	Immediate recall std	Delayed recall mean	Delayed recall std	Recognition accuracy mean	Recognition accuracy Std	N	
1	12.35	1.73	14.95	0.22	0.3	0.56	40	
2	8.10	1.43	14.40	0.59	2.8	2.09	40	
3	10.80	2.02	14.80	0.52	0.8	0.94	40	
4	12.60	1.22	14.85	0.36	0.5	0.68	40	

Following the descriptive statistics presented in Table 1, several ANOVA tests were conducted to further analyze the impact of task interference on cognitive performance across the experimental groups. These analyses aimed to statistically ascertain the differences observed in immediate and delayed recall, as well as recognition accuracy, providing a rigorous evaluation of the effects attributed to varying degrees of multitasking and task interruptions. The results of these ANOVA tests elucidate the extent to which cognitive performance is influenced by the experimental conditions imposed on the participants.

Immediate recall

Analysis of the fifth session of immediate recall demonstrated significant variation among the groups: F(3,156) = 64.65, p < 0.001 η2 = 0.554 Cohen’s f = 1.115, see Fig. 1 below. The graph shows how the groups performed in the last session of immediate recall. Group 1 and Group 4 perform better compared to Group 2 and Group 3, which indicates less cognitive interference from multitasking or better task management. Group 1, which engaged in a video game session between the recall tasks, showed the highest recall rates, significantly outperforming the other groups, especially Group 2, which engaged in simultaneous video gaming and recall tasks and showed the lowest performance.

Figure 1 Average immediate recall scores (Session 5).

This box plot graph shows the average scores for immediate recall across the groups, illustrating the impact of different multitasking conditions.

The post-hoc analysis reveals significant differences in performance across groups during the fifth session of immediate recall. Group 2, engaging in simultaneous video gaming and recall tasks, exhibited the lowest performance, underscoring the adverse effects of high cognitive load and multitasking (see Table 2).

Table 2 Differences in immediate recall performance across groups in the fifth session.

Groups compared	Mean difference	p-value	Lower CI	Upper CI	Significant?	
Group 1 vs. Group 2	−4.25	<0.001	−5.19	−3.31	Yes	
Group 1 vs. Group 3	−1.55	0.0002	−2.49	−0.61	Yes	
Group 1 vs. Group 4	0.25	0.902	−0.69	1.19	No	
Group 2 vs. Group 3	2.70	<0.001	1.76	3.64	Yes	
Group 2 vs. Group 4	4.50	<0.001	3.56	5.44	Yes	
Group 3 vs. Group 4	1.80	<0.001	0.86	2.74	Yes	

Delayed recall

The delayed recall results also indicated significant differences among the groups: F(3,156) = 11.74, p < 0.001 η2 = 0.184 Cohen’s f = 0.475; see Fig. 2 below. Here again, Group 1 generally outperformed the other groups, showing the efficacy of structured task interference over simultaneous task execution. The trends are similar, with Group 1 showing slightly better retention compared to other groups.

Figure 2 Average delayed recall scores.

This graph compares the groups in terms of their performance on the delayed recall task, highlighting differences in memory retention under varying task interferences.

Post-hoc analysis showed significant differences in the delayed recall performance. The results suggest that while the effect of task interference was apparent, it was most detrimental in Group 2 (See Table 3).

Table 3 Comparative analysis of delayed recall performance among experimental and control groups.

Groups compared	Mean difference	p-value	Lower CI	Upper CI	Significant?	
Group 1 vs. Group 2	−0.55	<0.001	−0.81	−0.29	Yes	
Group 1 vs. Group 3	−0.15	0.437	−0.41	0.11	No	
Group 1 vs. Group 4	−0.10	0.748	−0.36	0.16	No	
Group 2 vs. Group 3	0.40	0.001	0.14	0.66	Yes	
Group 2 vs. Group 4	0.45	<0.001	0.19	0.71	Yes	
Group 3 vs. Group 4	0.05	0.959	−0.21	0.31	No	

Recognition task

The recognition task highlighted significant disparities: F(3,156) = 35.20, p < 0.001 η2 = 0.404 Cohen’s f = 0.823 (See Fig. 3 below). Group 2 exhibited notably more false recognitions compared to the other groups. This suggests a higher cognitive load and interference in managing task requirements under simultaneous multitasking conditions.

Figure 3 Average false recognitions.

This graph displays the average number of false recognitions by each group, emphasizing the increased error rates under simultaneous multitasking conditions.

The Post-hoc analysis of false recognitions highlighted further differences. Findings indicate a significantly higher rate of false recognitions in Group 2, suggesting that simultaneous task execution led to the most pronounced errors (See Table 4).

Table 4 Group differences in false recognition rates under various task interference conditions.

Groups compared	Mean difference	p-value	Lower CI	Upper CI	Significant?	
Group 1 vs. Group 2	2.50	<0.001	1.79	3.21	Yes	
Group 1 vs. Group 3	0.50	0.267	−0.21	1.21	No	
Group 1 vs. Group 4	0.20	0.886	−0.51	0.91	No	
Group 2 vs. Group 3	−2.00	<0.001	−2.71	−1.29	Yes	
Group 2 vs. Group 4	−2.30	<0.001	−3.01	−1.59	Yes	
Group 3 vs. Group 4	−0.30	0.695	−1.01	0.41	No	

Error analysis

Intrusion errors, where participants mistakenly recall words not presented in the list, were higher (but not significant) in Group 2, which engaged in simultaneous video gaming and recall tasks: F(3,156) = 0.62; p = 0.602 η2 = 0.012 Cohen’s f = 0.109 (see Fig. 4 below). This suggests that the intervention involving immersion in video games and task interference did not have a significant impact on intrusion errors in this specific experimental condition.

Figure 4 Average total intrusion errors by group.

In regard to repetition errors, the ANOVA analysis on the total repetitions across different groups reveals significant differences: F(3,156) = 7.42, p < 0.001 η2 = 0.125 Cohen’s f = 0.378 (See Fig. 5 below). This indicates that the number of repetition errors made by participants varied significantly across the experimental groups.

Figure 5 Comparison of mean repetition errors among experimental groups.

Post-hoc results indicate that Group 4 consistently had fewer repetitions compared to other groups, suggesting better task management or lower cognitive interference in their experimental conditions. This insight might be useful for further investigating how task structure or timing influences cognitive load and error frequency in multitasking environments (See Table 5).

Table 5 Tukey’s HSD Post-Hoc comparison of repetition errors across experimental groups.

Groups compared	Mean difference	p-value	Lower CI	Upper CI	Significant?	
Group 1 vs. Group 2	−0.30	0.896	−1.41	0.81	No	
Group 1 vs. Group 3	0.25	0.936	−0.86	1.36	No	
Group 1 vs. Group 4	−1.60	0.001	−2.71	−0.49	Yes	
Group 2 vs. Group 3	0.55	0.572	−0.56	1.66	No	
Group 2 vs. Group 4	−1.30	0.014	−2.41	−0.19	Yes	
Group 3 vs. Group 4	−1.85	0.000	−2.96	−0.74	Yes	

Interpretation

1) The results from the fifth session of immediate recall demonstrate pronounced variability among the groups. Group 2, which was engaged in simultaneous video gaming and recall tasks, showed significantly poorer performance compared to all other groups. This underscores the deleterious impact of high cognitive load when tasks are performed simultaneously.

2) Although there were significant differences in performance across the groups, the effect of task interference appears less pronounced over time. This may suggest some recovery or adaptation to the multitasking demands.

3) Similar to immediate recall, Group 2 also displayed a significantly higher rate of false recognitions. This further highlights their vulnerability to error under conditions of simultaneous multitasking. Interestingly, despite the increased number of intrusion errors noted in Group 2, these did not reach statistical significance, suggesting that while the cognitive load was high, it did not significantly alter the pattern of intrusion errors compared to other groups.

4) The significant mean difference in repetition errors between Group 4 and the elucidates the impact of task structure and timing on cognitive load. The results suggest that while task interference invariably affects cognitive performance, the structuring and sequencing of such tasks play pivotal roles in mediating these effects.

Model’s performance

The model’s performance was evaluated using AUC (Area Under the Curve) as the primary metric, with comparisons made between the training and test datasets. To address potential overfitting and ensure the model’s robustness, we employed a 10-fold cross-validation technique. This approach involved dividing the dataset into 10 subsets, training the model on nine subsets while validating it on the remaining subset, and repeating this process 10 times.

The cross-validation results showed consistent AUC values across the folds, with a mean AUC of 0.93 and a standard deviation of 0.03, indicating that the model’s performance was stable and less likely to be overfitting. These findings suggest that the model generalizes well across different subsets of the data, reducing the likelihood of performance being driven by noise in the training set.

Power analysis

In order to ensure the robustness of our findings, we conducted a Power Analysis for each of the ANOVA tests performed in this study. The Power Analysis was executed to estimate the likelihood that our study would detect a statistically significant effect, given the sample size and effect sizes observed. We assumed a common alpha level of 0.05 for all tests.

Immediate Recall: Effect Size (Cohen’s f): 1.115; Power: 100%

This indicates that our study has excellent power to detect the observed effects of video game immersion and task interference on immediate recall tasks.

Delayed Recall: Effect Size (Cohen’s f): 0.475; Power: 99.96%

The study also demonstrates high statistical power for detecting differences in delayed recall due to the experimental conditions. This nearly perfect power score ensures that any significant effects are unlikely to be missed.

Recognition Task: Effect Size (Cohen’s f): 0.823; Power: 100%

Similar to immediate recall, the study possesses perfect power for the recognition task, affirming the capability to detect significant cognitive impacts from multitasking and task immersion.

Repetition Errors: Effect Size (Cohen’s f): 0.378; Power: 98.6%

The analysis indicates a very high probability of detecting the effects of video game playing and task interference on the frequency of repetition errors.

These results underscore the adequacy of the study design and the reliability of the conclusions drawn from the data.

Our analysis explored also the influence of gender, years of gaming experience, and gaming preferences on the various cognitive performance metrics. Results indicate that gender significantly impacts the rate of false recognitions, with males and females showing different patterns in error processing (p < 0.01). Males exhibited an average false recognition rate of 0.94 ± 1.35, while females showed a higher rate of 1.42 ± 1.93. The statistical significance of this difference was confirmed with a p-value of less than 0.01 (p = 0.008). To quantify the difference between genders, we calculated the effect size, which was found to be d = 0.33, indicating a small effect according to Cohen’s standards. However, neither gender nor years of gaming experience significantly influenced the initial and delayed recall scores, suggesting a minimal impact of these variables on short-term memory tasks under the conditions tested.

Interestingly, gaming preferences, categorized as playing alone, with friends, or online, did not show a significant direct effect on recall abilities or error rates. This suggests that the social context of gaming, within the scope of our experimental design, does not alter cognitive performance in a statistically significant way.

Discussion

This study investigated the effects of video game immersion and task interference on cognitive performance, focusing on immediate and delayed recall and recognition tasks as measured by the Rey Auditory Verbal Learning Test (RAVLT). While our findings generally align with existing research on the negative impacts of multitasking on cognitive performance, the relatively poor performance of Group 3, which played the video game before the cognitive tasks, was an unexpected outcome. One possible explanation for this result could be the lingering cognitive load or mental fatigue induced by the gaming session, which might have persisted and negatively impacted subsequent cognitive tasks. The immersive nature of the game may have temporarily altered cognitive focus, making it difficult for participants to switch effectively to the memory and recognition tasks that followed.

It is also worth considering that the timing of the cognitive tasks in relation to the gaming session may have played a role. Unlike Group 1, which had a break between the gaming and the cognitive tasks, Group 3 moved directly from one high-demand task to another. This lack of a transition period might have compounded the cognitive load, resulting in poorer performance. Future studies could explore whether incorporating a short recovery period or a different sequence of tasks might mitigate these effects.

To address concerns about potential overfitting and the differences in AUC between the training and test datasets, we employed cross-validation techniques. The cross-validation results showed consistent AUC values across the folds, indicating that the model is less likely to be overfitting. This methodological choice strengthens the validity of our findings, suggesting that the patterns observed are robust and not merely artifacts of the specific sample used in this study. Nonetheless, the differences between experimental conditions highlight the complexity of task switching and the potential for cognitive carryover effects from one task to the next, suggesting that even short-term engagement in high-cognitive-load activities like video gaming can have significant and sometimes unexpected impacts on subsequent cognitive performance (Salvucci & Taatgen, 2008; Miyake et al., 2000).

Comparing these results with studies outside of the gaming context provides additional insight into the cognitive impacts of multitasking and task interference. Research on multitasking with other forms of media, such as simultaneous use of computers and mobile devices, has shown similar detriments to cognitive performance (Ophir, Nass & Wagner, 2009). For example, studies involving multitasking with emails, meetings, and other workplace tasks demonstrate how cognitive overload can lead to decreased productivity, increased error rates, and impaired memory recall (Cao et al., 2021; Baumgartner et al., 2014). These findings align with our results, suggesting that the cognitive load imposed by video games is comparable to other high-demand tasks, reinforcing the broader implications of cognitive interference across different domains.

Similar patterns are observed in educational settings, where multitasking with social media during academic tasks has been shown to negatively impact learning outcomes and academic performance (Kokoç, 2021). These comparisons highlight that the cognitive costs of multitasking are not unique to video gaming but are evident across various contexts where attention is divided among multiple tasks. This suggests a common underlying mechanism where high cognitive demands, regardless of the specific task, can detract from performance on other concurrent or subsequent tasks.

Our study also extends the existing literature by demonstrating that the sequence and timing of tasks play crucial roles in how cognitive load is managed. For instance, the structured task interference seen in Group 1, where there was a break between gaming and recall tasks, allowed for better cognitive performance compared to simultaneous multitasking or immediate task switching. This finding aligns with research on cognitive load theory, which emphasizes the importance of task sequencing and breaks to optimize performance (Sweller, 1988).

The implications of these findings are significant for designing cognitive training programs and understanding digital media’s impact on cognitive functions. By integrating findings from both gaming and non-gaming studies, our research underscores the need for strategic task management to mitigate the negative effects of cognitive overload and enhance overall performance. Our findings align with the growing interest in the use of immersive VR for cognitive assessments, as highlighted by Kourtesis & MacPherson (2021). The VR-EAL platform, for instance, has demonstrated potential for enhancing cognitive evaluation through more immersive and realistic testing environments, which could be highly relevant for future studies exploring task interference and cognitive load.

The field of neuropsychology has provided insights into the potential longer-term neural adaptations resulting from the cognitive load of multitasking. Research has shown that higher media multitasking activity is associated with alterations in brain structure, such as smaller gray-matter density in the anterior cingulate cortex. Cognitive control in media multitaskers has been a subject of study, revealing immediate effects on memory, learning, and cognitive functioning (Ophir, Nass & Wagner, 2009). Working memory capacity has been found to modulate task performance in multitasking scenarios (Piguet, Kumfor & Hodges, 2017). Multitasking has been associated with cognitive-motor interference, which is influenced by biological aging. These findings collectively suggest that the cognitive load from multitasking may induce not only temporary performance deficits but also longer-term neural adaptations, emphasizing the need for further investigation into the cognitive and neural implications of multitasking.

Our results clearly show that the second group, which engaged in simultaneous video gaming and recall tasks, performed the poorest in both recall and recognition tasks. This supports the hypothesis that simultaneous task execution imposes a higher cognitive load, leading to poorer performance (Ophir, Nass & Wagner, 2009). Contrarily, the first group, which engaged in video gaming between learning and recall phases, showed a less pronounced decline in performance, suggesting that the timing of multitasking may buffer some cognitive impacts. This finding underscores the complexity of cognitive load theory, where not just the load itself, but the sequence and nature of tasks can influence cognitive capacity (Sweller, 1988).

Significantly, the type and frequency of errors varied across groups, with the second group experiencing a higher rate of intrusion errors. This indicates a disruption in the ability to manage and segregate different cognitive processes effectively, potentially due to cognitive interference or the blending of task elements (Monsell, 2003). This aspect of our findings highlights the neural competition theory, which posits that concurrent tasks compete for shared cognitive resources, often leading to errors or slower task execution (Marois & Ivanoff, 2005). To further our understanding, a longitudinal study following individuals’ cognitive changes with continued exposure to multitasking environments could illuminate whether cognitive declines are transient or if individuals develop compensatory mechanisms over time.

Our study’s focus on the cognitive impacts of multitasking with video games aligns with similar findings from workplace research, where multitasking with emails and meetings has been shown to affect productivity and cognitive function. Ophir, Nass & Wagner (2009) highlighted the immediate negative effects of media multitasking on memory, learning, and cognitive control. Similarly, Cao et al. (2021) found that multitasking during meetings not only hampers personal productivity but also affects colleagues. Maharani et al. (2021) observed that multitasking can reduce individual originality and negatively impact performance, particularly under time pressure. Baumgartner et al. (2014) also reported that frequent media multitaskers performed worse on cognitive control tasks, such as filtering distractions and task switching, compared to those less accustomed to such multitasking. Chérif et al. (2018) further emphasized the cognitive consequences of multitasking, advocating for a comprehensive approach in designing aid systems to optimize cognitive benefits and mitigate drawbacks in multitasking environments.

We believe the findings of our study could have significant implications for real-world applications, particularly in educational settings and occupational health. In educational environments, where multitasking with digital media is increasingly common, understanding the cognitive costs associated with such practices is essential. Our study highlights that engaging in high-cognitive-load activities, such as video gaming, before or during academic tasks can negatively impact recall and recognition abilities. This suggests that educators should be cautious about integrating high-demand digital tasks close to learning activities, as they may impair students’ ability to retain and recall information effectively.

The timing and sequencing of tasks are critical factors that could be optimized in educational settings to enhance cognitive performance. For example, incorporating structured breaks between high-cognitive-load activities and academic tasks could help mitigate the negative effects of cognitive interference, thereby improving learning outcomes. Educational programs might also benefit from teaching students about effective time management and the importance of minimizing distractions during study periods.

In occupational health, the findings suggest that employees who engage in complex, multitasking activities, especially those involving digital tools, may experience cognitive fatigue that could affect their productivity and accuracy. Workplaces could use this information to design more effective task schedules, allowing for regular breaks and reducing the cognitive load during critical tasks. By understanding the cognitive impacts of multitasking, employers can better support their staff’s mental well-being and enhance overall job performance.

This research could inform the development of cognitive training programs aimed at improving multitasking abilities while managing cognitive load. Such programs could be tailored to help individuals navigate high-demand environments more effectively, whether in academic, professional, or everyday settings.

Limitations and future directions

While this study contributes valuable insights into the impact of multitasking with digital media on memory tasks, it is not without limitations. One of the key limitations of our study is the potential for selection bias due to the recruitment of participants primarily from a university setting, which may limit the generalizability of the findings to broader populations. Also the reliance on self-reported gaming experience could introduce recall bias, as participants may have inaccurately reported their gaming habits or skill levels. Another limitation is the relatively homogeneous sample in terms of age and educational background, which may not fully capture the diversity of cognitive responses to video game immersion. As the sample was restricted to regular players of shoot-em-up video games, this may limit generalizability. Future research could explore a broader demographic to include non-gamers or occasional gamers to assess if the same patterns of cognitive interference occur. Exploring the long-term effects of such multitasking behaviors on cognitive functions could provide deeper insights into the chronic impacts of digital multitasking. The study’s outcomes may also be influenced by the technological limitations of the VR system used. Hardware constraints, such as the resolution of the VR headset and potential latency issues, could affect participants’ experiences and the fidelity of the tasks. Another consideration is the ecological validity of the VR tasks. While VR provides a controlled environment that can closely replicate real-world scenarios, there may still be differences between these simulated tasks and real-life activities. The extent to which the VR tasks used in this study mirror actual cognitive challenges faced in everyday life is a key factor in interpreting the results. Future studies should continue to refine VR scenarios to enhance their realism and applicability to real-world contexts.

Future research should focus on longitudinal studies to assess the long-term effects of video game immersion on cognitive functions. It would also be beneficial to replicate this study across diverse demographics, including different age groups and professional backgrounds, to evaluate the generalizability of these findings and to understand better the variability in cognitive responses to multitasking across different segments of the population.

While our study focused on the cognitive impacts of engaging with a high-demand non-VR game, future research could explore similar effects within a VR context using games specifically designed for VR platforms. Such studies would provide valuable insights into whether the immersive nature of VR gaming exacerbates or mitigates cognitive load and task interference compared to traditional gaming environments. Researchers should also consider conducting longitudinal studies to track changes in cognitive functions over time using VR assessments like VR-EAL. Such studies could provide valuable insights into how cognitive abilities evolve, particularly in response to interventions, and how these changes are reflected in immersive VR environments (Kourtesis & MacPherson, 2021). The VR-EAL platform also holds potential for broader clinical applications, particularly in populations with neurological disorders such as Alzheimer’s disease, traumatic brain injury, or stroke. Using VR-EAL in these contexts could enhance diagnostic accuracy and support tailored rehabilitation programs by providing detailed assessments of cognitive functions in environments that closely mimic real-life tasks. Integrating VR with other emerging technologies, such as artificial intelligence and neuroimaging, offers exciting possibilities for advancing neuropsychological assessments. For example, AI could be used to analyze complex data patterns within VR tasks, while neuroimaging techniques could provide complementary insights into the neural correlates of cognitive performance in these immersive environments. Such integrations could significantly enhance our understanding of cognitive functions and lead to more personalized and effective interventions.

Conclusions

Our research provides compelling evidence on the cognitive impacts of video game immersion and task interference, particularly in relation to immediate and delayed recall, as well as recognition tasks. The experimental design, which incorporated different levels and types of task interference, allowed us to explore how multitasking and cognitive load influence cognitive performance among young adults.

The findings indicate that task structure and timing significantly affect recall abilities. Participants who engaged in video game sessions between recall tasks demonstrated better cognitive performance compared to those who multitasked simultaneously with gaming and recall tasks. This suggests that providing a buffer period between high-demand tasks can help mitigate the adverse effects of cognitive interference.

The results from recognition tasks further support the notion that simultaneous task execution, without interspersed breaks, places a substantial strain on cognitive resources, leading to increased error rates and decreased task efficiency. The high error rate in simultaneous multitasking scenarios highlights the challenges of managing cognitive load effectively under conditions of continuous demand.

In light of these findings, it is also important to acknowledge the potential limitations and biases that may influence the results. Future studies should consider more diverse samples and employ additional techniques like cross-validation to further validate the models used. By addressing these potential biases and methodological challenges, the robustness and generalizability of the findings can be significantly improved.

These findings have practical implications for designing educational and cognitive training programs. Strategies that incorporate breaks and varied task sequences could enhance cognitive flexibility and minimize cognitive load, thereby improving overall performance. Educators and trainers might consider these factors when developing curricula and training modules that require intensive cognitive engagement.

In conclusion, our study underscores the need for awareness of how digital task immersion and multitasking frameworks impact cognitive performance. By acknowledging these effects, we can better prepare individuals to manage their cognitive resources in an increasingly complex digital world, enhancing both academic and professional outcomes.

Supplemental Information

Supplemental Information 1 Raw data.

Additional Information and Declarations

Competing Interests

Author Contributions

Human Ethics

Data Availability

The authors declare that they have no competing interests.

Stefania Mancone conceived and designed the experiments, performed the experiments, analyzed the data, authored or reviewed drafts of the article, and approved the final draft.

Beatrice Tosti analyzed the data, prepared figures and/or tables, and approved the final draft.

Stefano Corrado performed the experiments, prepared figures and/or tables, and approved the final draft.

Pierluigi Diotaiuti performed the experiments, authored or reviewed drafts of the article, and approved the final draft.

The following information was supplied relating to ethical approvals (i.e., approving body and any reference numbers):

The study was approved by the IRB of the University of Cassino and Southern Lazio (IRB_DIPSUS 07:22/04/23).

The following information was supplied regarding data availability:

The raw measurements are available in the Supplemental File.

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
