# Peer review of "Effects of video game immersion and task interference on cognitive performance: a study on immediate and delayed recall and recognition accuracy"

_PeerJ, doi:10.7717/peerj.18195_

## Round 0.1 · original submission · Major Revisions

Having read the manuscript, I do agree with the issues raised by the Reviewers. Nonetheless, I believe the manuscript fills an important gap in the literature and adds valuable insights to the field. As such, I'd ask the Authors to address all the issues raised by the reviewers. For instance, I think R1 is right in suggesting to be more concise in the introduction, since it is a little bit redundant. R1 also suggests to include more recent references, and the suggestion made by R3 might help in addressing this issue.

Thank you for submitting this manuscript to PeerJ. I'm looking forward to receiving the revised version.

·

Basic reporting

This study investigates the cognitive impacts of video game immersion and task interference on immediate/delayed recall and recognition tasks. I believe the study to be well-designed and conducted, and the manuscript to be generally well-written, although it is at times redundant. While the core hypothesis of the study is rather straightforward, the three different experimental groups add a bit more information. I only have minor comments, detailed below.

Experimental design

Solid.

Validity of the findings

Adequate (although the poor performance of group 2 is certainly not unexpected)

Additional comments

Minor comments:

The manuscript would benefit from more concise language, particularly in sections like the introduction and methodology where certain points are repeated. For example, the rationale for choosing Call of Duty is mentioned multiple times and could be summarized in one paragraph.

Some references appear a bit outdated, though likely included for historical reasons.

The discussion describes the results' implications adequately. However, I was a bit surprised by the relatively poor performance of Group 3, which played before the task. Maybe the discussion could spend more time on this (for me) unexpected result.

The discussion could be expanded to consider implications for cognitive training and digital media use.

Reviewer 2 ·

Basic reporting

Strengths: The manuscript is well-structured, written in clear, professional English, and provides a detailed background on cognitive impacts of video gaming and task interference. It cites relevant literature effectively.

Areas for Improvement: The introduction could be expanded to provide a clearer justification for the study’s focus on specific video game genres and cognitive tasks. There are minor grammatical issues that need refinement to enhance clarity, particularly in the methodology description (Lines 110-140).

Experimental design

Strengths: The design is rigorous, with clearly defined and relevant research questions. The study fills a gap in understanding how video game immersion impacts cognitive performance in multitasking environments.

Areas for Improvement: The methods section should detail the statistical power of the tests used, which would help in understanding the robustness of the study findings. Clarification on the randomization process in group assignments could enhance the transparency of the study design.

Validity of the findings

Strengths: The findings are robust, supported by detailed statistical analyses (ANOVA and post-hoc tests), and align well with the hypothesis. The study uses a large sample size, improving the reliability of the results.

Areas for Improvement: There is a need for a deeper discussion on the potential biases and limitations of the study. The differences in AUC between training and test datasets suggest possible overfitting; addressing this with cross-validation techniques could strengthen the validity of the model.

Additional comments

The paper provides insightful data on the cognitive costs associated with multitasking involving video games. However, the discussion could benefit from comparisons with existing studies, particularly those not involving video gaming, to broaden the context of the findings.
It would be helpful to include a more detailed discussion on the implications of these findings for real-world applications, such as educational settings or occupational health.
The manuscript would benefit from a thorough proofreading by a native English speaker to polish the language and improve the overall readability.

·

Basic reporting

Clear and unambiguous, professional English used throughout:
The manuscript is well-written with clear, professional English throughout. However, some sentences are long and complex, which might affect readability. Consider revising sentences like, "The recruitment students for the study was conducted a voluntary basis..." to "The recruitment of students for the study was conducted on a voluntary basis."

Literature references, sufficient field background/context provided:
The manuscript is well-referenced with a comprehensive background. However, including more recent studies on VR and neuropsychology would enhance the context. See Kourtesis & MacPherson, 2021 for an overview and relevant refs.

Kourtesis, P. & MacPherson, S. E. (2021). How immersive virtual reality methods may meet the criteria of the National Academy of Neuropsychology and American Academy of Clinical Neuropsychology: A software review of the Virtual Reality Everyday Assessment Lab (VR-EAL). Computers in Human Behavior Reports, 4, 100-151 https://doi.org/10.1016/j.chbr.2021.100151

Professional article structure, figures, tables. Raw data shared:
The article follows a professional structure with clear sections. Figures and tables are relevant but could be improved using box plots or violin plots for better visualization of data distribution. The raw data is shared, meeting transparency requirements.

Self-contained with relevant results to hypotheses:
The manuscript is self-contained, presenting relevant results that address the hypotheses. The conclusions are well-supported by the data.

Experimental design

Original primary research within Aims and Scope of the journal:
The research is original and falls well within the journal's aims and scope.

Research question well defined, relevant & meaningful:
The research questions are well-defined, relevant, and meaningful, clearly articulating how the work fills a gap in the literature.

Rigorous investigation performed to a high technical & ethical standard:
The investigation is thorough and conducted to high technical and ethical standards, with ethical considerations properly addressed.

Methods described with sufficient detail & information to replicate:
The methods section is detailed, providing sufficient information for replication. However, more details on the statistical methods, particularly handling missing data and outliers, would enhance clarity.

Validity of the findings

Impact and novelty not assessed. Meaningful replication encouraged where rationale & benefit to literature is clearly stated:
The manuscript appropriately focuses on the validity of the findings rather than impact and novelty. The rationale for replication is well-stated, and the benefits to the literature are clear.

All underlying data have been provided; they are robust, statistically sound, & controlled:
The data provided are robust and statistically sound. However, a more detailed discussion on handling outliers and missing data would strengthen the manuscript.

Conclusions are well stated, linked to original research question & limited to supporting results:
The conclusions are well-stated, directly linked to the original research questions, and supported by the results.

Additional comments

Introduction Enhancement:
The introduction could benefit from a stronger opening statement to engage readers immediately. For example, start with a compelling statistic or a real-world example that underscores the significance of using VR in neuropsychological assessments.

Figures Improvement:
Consider using box plots or violin plots for figures to improve data visualization. Box plots provide a clear summary of data distribution, highlighting medians, quartiles, and potential outliers. Violin plots add a layer of density estimation, showing the distribution shape, which can be particularly useful for illustrating the data's distribution more comprehensively.

Discussion of Limitations:
Expand the discussion on potential limitations of the study:

Generalizability: Address how the findings might differ in diverse populations or settings. Mention if the sample was limited in terms of demographics (e.g., age, gender, socioeconomic status) and how this might affect the results.
Technological Limitations: Highlight any limitations related to the VR technology used, such as hardware constraints or software limitations, and how these might impact the study's outcomes.
Ecological Validity: Discuss the extent to which the VR tasks replicate real-world tasks and the implications for ecological validity.
Future Directions:
Emphasize future research directions and potential clinical applications:

Longitudinal Studies: Suggest conducting longitudinal studies to examine how cognitive functions assessed in VR might change over time or in response to interventions.
Broader Applications: Discuss the potential for using VR-EAL in various clinical populations, such as those with neurological disorders, and how it might aid in diagnosis and rehabilitation.
Integration with Other Technologies: Consider the integration of VR with other emerging technologies, such as artificial intelligence or neuroimaging, to enhance the assessment and understanding of cognitive functions.
Technological Considerations:
Ensure that the technological aspects of VR implementation are well-documented:

User Experience: Discuss how user feedback was incorporated to improve the VR tasks. This includes usability testing and adjustments made based on participants' experiences.
Technical Details: Provide more detailed information on the VR hardware and software used, including specifications and setup. This will help other researchers replicate the study accurately.

---

## Round 0.2 · accepted · Accept

Dear Authors,

All the original Reviewers have reviewed the revised manuscript and they all agree in the decision of accepting it, as they all point out that you have addressed all their previous concerns. So it is my pleasure to inform to tell you that your manuscript have been accepted for publication.

·

Basic reporting

-

Experimental design

-

Validity of the findings

-

Additional comments

The authors responded to my suggestion satisfactorily.

Reviewer 2 ·

Basic reporting

The manuscript has been modified regarding previous suggestions, I'd like to recommand accepting it.

Experimental design

The manuscript has been modified regarding previous suggestions, I'd like to recommand accepting it.

Validity of the findings

The manuscript has been modified regarding previous suggestions, I'd like to recommand accepting it.

·

Basic reporting

No comment

Experimental design

No comment

Validity of the findings

No comment

Additional comments

The authors satisfactorily addressed my comments. I believe that the manuscript is now of adequate quality for publication.